# Zinc in Human Health and Infectious Diseases

**DOI:** 10.3390/biom12121748

**Published:** 2022-11-24

**Authors:** Martina Maywald, Lothar Rink

**Affiliations:** Institute of Immunology, Faculty of Medicine, RWTH Aachen University Hospital, 52074 Aachen, Germany

**Keywords:** zinc, zinc homeostasis, zinc transporter, infectious diseases, pro- and anti-inflammatory signaling pathways

## Abstract

During the last few decades, the micronutrient zinc has proven to be an important metal ion for a well-functioning immune system, and thus also for a suitable immune defense. Nowadays, it is known that the main cause of zinc deficiency is malnutrition. In particular, vulnerable populations, such as the elderly in Western countries and children in developing countries, are often affected. However, sufficient zinc intake and homeostasis is essential for a healthy life, as it is known that zinc deficiency is associated with a multitude of immune disorders such as metabolic and chronic diseases, as well as infectious diseases such as respiratory infections, malaria, HIV, or tuberculosis. Moreover, the modulation of the proinflammatory immune response and oxidative stress is well described. The anti-inflammatory and antioxidant properties of zinc have been known for a long time, but are not comprehensively researched and understood yet. Therefore, this review highlights the current molecular mechanisms underlying the development of a pro-/ and anti-inflammatory immune response as a result of zinc deficiency and zinc supplementation. Additionally, we emphasize the potential of zinc as a preventive and therapeutic agent, alone or in combination with other strategies, that could ameliorate infectious diseases.

## 1. Introduction

The essential trace element zinc is extraordinarily important for a well-functioning immune system and for human health in general. Zinc has a key role in several biological processes of the human body, such as influencing cell differentiation, proliferation, and apoptosis, which affect the growth of the organism [1]. The importance of zinc quickly becomes apparent when one considers bioinformatics studies that have located nearly 3000 human proteins that are presumed to bind zinc [2]. Of these, zinc fingers and zinc finger-containing domains in particular require the divalent cation as a stabilizer. Others need zinc for catalytic and regulatory functions, as it is needed for the activity of several metalloenzymes [2,3]. It is well established that zinc is the second most abundant micronutrient in the human body after iron [4,5]. Since the first discovery of zinc being important for human health [6], much research has been performed. Nowadays it is known that zinc, and especially zinc deficiency, plays an important role in many metabolic and chronic diseases, including: diabetes, cancer (e.g., esophageal, hepatocellular, breast cancer, colon cancer), neurodegenerative diseases such as Alzheimer’s disease, and intestinal diseases such as inflammatory bowel disease and irritable bowel syndrome [7,8,9,10,11,12,13]. There is also strong evidence between zinc deficiency and several infectious diseases such as malaria, HIV, tuberculosis, measles, and respiratory infections such as COVID-19 pneumonia [14,15,16]. Consequently, zinc’s myriad of functions in the human immune system demonstrates how a defect in nutritional absorption may lead to the manifestation of various diseases. Nevertheless, both zinc deficiency and oversupply affect homeostasis, and consequently, cellular function. Zinc intoxication is a rare event; however, it was seen by using, e.g., denture adhesive creams that contain high amounts of zinc. Here, high amounts of zinc were reported to cause nausea, dizziness, headaches, gastric distress, vomiting, and loss of appetite [17,18]. If used for weeks, doses of 50 mg zinc or more can interfere with copper absorption, leading to a low copper status, which consequently impairs immune function [16,17,18,19]. Hence, zinc homeostasis has to be carefully monitored and regulated.

This review highlights the particularly important role of zinc for a well-functioning immune system and its versatile influence on various signaling pathways of the pro- and anti-inflammatory immune response. On the one hand, it is shown how inadequate nu-trient intake can lead to the occurrence of various diseases caused by harmful pathogens such as bacteria, fungi, or viruses. On the other hand, the impact of zinc supplementation strategies that can lead to faster remission or alleviation of disease symptoms is high-lighted, illustrating the therapeutic benefits of zinc in various infectious diseases.

Search strategy: A comprehensive search of the databases Web of Science, PubMed, and Cochrane Library was conducted. The search was conducted using a combination of keywords including (1) “zinc” OR “zinc supplementation”, AND (2) “infectious diseases”, “viral disease” “bacterial diseases”, “pneumonia”, “common cold”, “hepatitis”, “HIV”, “dengue fever”, “COVID-19”, “tuberculosis”, “shigellosis”, “H. pylori”, “diarrhea”, “malaria”, “leishmaniosis”, “fungi”, or “fungal diseases”, and related terms. We also manually searched references of relevant literature to identify other eligible sources.

## 2. Assessment of Zinc Status

Nowadays, there are a variety of methods to determine zinc status, which differ in terms of whether the question is scientific or diagnostic. Diagnostic zinc concentrations are determined as standard in whole blood, serum, plasma, or urine. For this purpose, the methods of atomic absorption spectroscopy (AAS, flame or graphite furnace), inductively coupled plasma optical (atomic) emission spectrometers (ICP-OES/ICP-AES), and ICP mass spectrometers (ICP-MS) are used often, providing excellent sensitivity, precision, and good accuracy with virtually no restriction of the element investigated [20,21].

The scientific approach is far more complex, as both the distribution and dynamics of zinc are important for biological processes that need to be specifically analyzed. Nowa-days, a growing number of fluorescent small-molecule probes and protein-based sensors exist that enable the measurement of subcellular zinc distribution in different organelles. Most of the available tools can be divided into two general classes of probes: small-molecule probes and protein sensors based on Förster resonance energy transfer (FRET). Small-molecule probes usually increase in fluorescence upon chelation of zinc, as known for the indicators Zinpyr-1, FluoZin-3, or Zinquin. They can be cell-permeable, and are therefore easy to apply. They provide a rapid means of investigating metal/protein equilibria by optical-sensing methods with both high sensitivity and very little sample preparation. They have been adapted to provide ratiometric signals that allow for the normalization of fluorescence changes that are not due to the chelation of zinc [22,23]. FRET consists of two fluorescent proteins and a zinc-coordinating site that is designed to change the relative orientation and distance of the fluorescent proteins upon binding to a target, which leads to a change in FRET signal. The sensors are genetically encoded and can be targeted to different organelles, as known for the Zap, eZinCh, or eCALWY series.

## 3. Zinc Intake and Distribution

Zinc needs to be properly distributed into all compartments of the human body to fulfill a multiplicity of functions. It is undisputed that there must exist a complex process for homeostasis and to ensure proper zinc allocation, because the zinc content between various organs differs greatly. Prostate, pancreas, and bone are considerably high in zinc, containing about 100 to 250 µg/g. Zinc concentrations in heart, brain, and plasma are comparatively low, at 1 to 23 µg/g. Although plasma only has a zinc content of about 1 µg/g, it is probably the most important reservoir for zinc homeostasis [24].

In total, the body of an adult human contains about 2.6 g of zinc. The largest proportion is found in bones and skeletal muscles (~86%), followed by skin (4.2%) and liver (3.4%). It should be noted that some zinc-containing entities, including the thymus and mucous membranes, were not included in the calculation, so the actual total zinc content of the human body is higher than the 2.6 g estimated [25,26]. Within body fluids, free zinc is rare, since it is predominantly bound to proteins, including albumin, α2 macroglobulin (A2M), and transferrin. Albumin binds zinc with a relatively low affinity, A2M with a medium affinity, and transferrin with a high affinity. However, although albumin’s binding affinity is low, hypoalbuminemia can even result in zinc deficiency [27].

Intracellularly, zinc is distributed mainly between specific zinc-storage structures called zincosomes (~50%) and the nucleus (~30–40%). The remaining zinc is distributed between the cytoplasm and other cell organelles [28,29]. In the cytoplasm, zinc is largely bound by zinc-chelating proteins called metallothioneins (MT). Those were first discovered in 1957 [30] and play an important role in zinc homeostasis by complexing approximately 20% of intracellular zinc acting as zinc buffer [31]. The availability of cytosolic zinc is mediated by zinc storage or release from organelles (e.g., zincosomes, endoplasmic reticulum (ER), or Golgi apparatus). Zincosomes are vesicle-like storage granules for zinc that are typically visualized with zinc-binding probes Zinquin or FluoZin-3 [32,33]. These granules have different functions depending on the cell type, such as zinc loading of milk, neuron function, and insulin secretion. Influx into the cell or efflux into the extracellular space is another important aspect in zinc availability [34], which is mediated by specific zinc-transporting proteins, such as zinc importers or exporters, and membrane channels, as described later [31,35].

During evolution, plant-, animal-, and human-based cells have developed metal-binding proteins to maintain low cytosolic metal ion concentrations and thus protect themselves from high cytotoxic metal concentrations. Interestingly, the human body evolved mechanisms to compartmentalize and sequester zinc, for example, to deny vital zinc to invading microorganisms. In contrast, intracellular release of high zinc amounts, for instance from proteins and organelles, may be beneficial and desirable because transiently and locally high zinc levels also target gene expression, enzymatic activity, and cell signaling of pathogens, leading to zinc intoxication [35,36].

To maintain healthy zinc homeostasis, the body’s own zinc loss needs to be compensated. Since there is no dedicated compartment for zinc storage in the human body, the trace element must be ingested daily in sufficient quantities with food [37]. Zinc loss can be of intestinal and nonintestinal origin, including fecal zinc excretions and excretions with urine, sweat, menstrual flow, and semen, as well as loss of hair, nails, and desquamated skin [38]. In addition, special life circumstances must be taken into account in which the presumed zinc loss must be individually adjusted, such as pregnancy, lactation, or even early infancy or young children in growth [38].

Zinc homeostasis in the human body is primarily regulated by absorption in the intestine, particularly in the duodenum and jejunum [39,40,41]. In this process, zinc transporters on the apical and basolateral membrane of enterocytes and the zinc-binding proteins (metallothioneins) regulate zinc homeostasis in the intestine and eventually in the human body [3,42]. Zinc absorption not only depends on adequate dietary intake, but is also strongly influenced by intestinal availability from food, which means that zinc bioavailability depends on dietary composition [43,44]. So far, it is known that foods prepared from unrefined cereals or legumes, or foods containing indigestible plant ligands such as phytate and lignin, form an insoluble complex with zinc, making zinc unavailable for resorption in the intestine. Lignin is important for plant structure and resilience, whereas phytate is the storage form of phosphorus in plants that binds and stores minerals. Hence, both phytate and lignin reduce the bioavailability of zinc because they bind zinc very effectively and inhibit its absorption, respectively [17,18,45,46]. Besides plant products or components, zinc absorption is also influenced by additional factors such as other trace element concentrations such as copper or iron, or mineral concentrations such as calcium, as mentioned later.

The highest zinc content is found in the following foods: red meat, offal, oysters and shellfish, fortified cereals, and whole-grain products [17,46]. Oysters contain more zinc per serving than any other food. However, in Western countries, beef contributes 20% of the zinc intakes from food because it is commonly consumed [47]. Eggs and dairy products also contain zinc [17]. Beans, nuts, and whole grains contain zinc, but the bioavailability is much lower because plant-based foods contain high amounts of phytates [17,18,46]. In summary, zinc is present in a wide variety of food classes, but its availability to the human body varies greatly. In general, zinc from animal sources has a higher bioavailability compared to zinc from plant products. People who consciously avoid eating red meat and eat mainly plant-based foods, such as vegetarians and vegans, but also people who often have to abstain from meat, such as the population living in developing countries, have a higher risk of developing zinc deficiency due to insufficient zinc intake [48].

The recommended dietary allowance (RDA) for daily zinc intake depends on various factors, such as age, gender, weight, and phytate content of the diet. The recommended levels vary slightly from country to country: the National Institute of Health recommends an intake of 11 mg/day for adult men and 8 mg/day for adult women [18]. During pregnancy and lactation 11 mg/day to 12 mg/day are recommended. For infants and children the RDA is 3 mg/day for both sexes [18]. The German Nutrition Society (DGE) recommends 11–16 mg/day for adult men and 7–10 mg/day for adult women, depending on phythate intake. During pregnancy and lactation up to 13–14 mg/day are recommended. For infants and children the RDA is 2.5 mg/day for both sexes [49]. Due to the inhibitory effect of phytate on zinc absorption from food, phytate levels are taken into account by both the World Health Organization (WHO) and the European Food Safety Authority (EFSA) when establishing recommendations for daily zinc intake. However, there are different classifications of the WHO and the EFSA: WHO divides diets into three groups according to their potential absorption capacity of zinc per phytate-zinc molar ratio: high (<5), moderate (5–15), and low (>15) zinc bioavailability [50]. EFSA and (DGE) provides several zinc reference recommendations for diets with phytate intakes of about 300–(330), 600–(660), 900–(990), and 1200 mg/day [49,51]. Oral zinc supplements are readily available, but not all offer the same zinc bioavailability. Zinc bound to amino acids such as aspartate, cysteine, and histidine shows the highest absorption concentration, followed by zinc chloride, sulfate, and acetate, whereas zinc oxide shows the lowest bioavailability [52].

The importance of zinc homeostasis is underlined by the extraordinarily high number of diseases associated with changes in zinc content in human organism. Most of these diseases are associated with zinc deficiency [24,53,54]. About two billion people worldwide suffer from zinc deficiency, which has serious consequences for their health, affecting every solid organ in the human organism as well as the immune system [6,24,55]. Nowadays, it is known that zinc deficiency is very common worldwide, especially in developing countries. Here, it is the fifth leading cause of loss of healthy life years [56]. In Western countries, the elderly population is particularly affected by zinc deficiency, as nearly 30% of them are considered to be zinc-deficient [24,57]. Moreover, vegetarian or vegan people are also concerned [58,59]. In general, the WHO estimates that about one-third of the world’s population is at risk for zinc deficiency [56]. Globally, it is responsible for approximately 16% of lower respiratory tract infections, 18% of malaria, and 10% of diarrheal diseases. While severe zinc deficiency is rare, mild-to-moderate deficiency is more common worldwide [54,56]. However, the lack of a suitable biomarker for physiological zinc status, and thus the poor ability to detect inadequate zinc absorption, especially in the early stages of mild zinc deficiency, is a major obstacle in this situation [48].

## 4. Regulation of Zinc Homeostasis

The distribution of zinc between the cytosol and cell organelles is mediated by specific zinc-transporting proteins, such as zinc importers or exporters, or by membrane channels. The mammalian zinc transporters belong to two major protein families, which are distinguished according to their specific membrane topology: Two major zinc transporter families, the SLC39 (ZIP) and SLC30 (ZnT) family, control cellular zinc homeostasis.

The first group includes 14 zinc transporters of the SLC39As family Zrt (zinc-regulated transporter)-like and Irt (iron-regulated transporter)-like proteins, called ZIP transporters. ZIPs 1-14 (SLC39A1-14) are abundantly located at the outer plasma membrane and at intracellular organelles and are responsible for transporting zinc both from the extracellular space and from intracellular compartments into the cytosol to increase cytoplasmic zinc.

The second group includes 10 zinc transporters of the SLC30As family (ZnT 1–10 (SLC30A1-10)) that decrease cytoplasmic zinc by transporting zinc either out of the cell into the extracellular space or into intracellular organelles such as Golgi apparatus, or zincosomes [60,61,62]. Both ZIP and ZnT transporters are expressed in a tissue-specific manner and are expressed in a variety of different cell types in the human body. They are localized to the outer plasma membrane and to organelle membranes such as mitochondria, Golgi apparatus, lysosomes, and ER. In addition, they show developmental and stimulus-dependent expression patterns, highlighting the need for adequate zinc levels for specific cellular functions [1,62].

More recently, zinc distribution has also been described in the context of receptors and ion channels other than ZIP or ZnT. These include, for example, nicotinic acetylcholine receptors, glutamatergic receptors, voltage-gated calcium channels, and transient receptor potential channels. Amino acid-bound zinc diffusion is also known, but these mechanisms remain to be further defined [1].

Interestingly, there are many zinc transporter mutations reported, to be involved in inherited diseases:

(1) Zinc importer: Mutation in the intestinal ZIP4 transporter causes the lethal autosomal-recessive inherited zinc deficiency disease, acrodermatitis enteropathica (AE). AE is characterized amongst other symptoms by gastrointestinal disturbances, retardation, and high susceptibility to infections. A complete cure can be achieved through high-dose zinc supplementation [63]. Mutation in ZIP 7 is associated with Agammaglobulinemia due to impaired B cell function and proliferation. Mutation in ZIP13 cause a novel subtype of Ehlers–Danlos Syndrome (EDS), where collagen synthesis and modification is impaired [64]. ZIP 14 mutation is associated with hypermanganism and Parkinson syndrome [65]. Additionally, various ZIP transporter modifications are addressed to different cancer types, such as ZIP 6 (esophageal squamous cell carcinoma and gastric adenocaricinoma), ZIP 9 (prostate cancer and breast cancer), and ZIP 12 (colorectal cancer and bladder cancer) [66].

(2) Zinc exporter: Mutation in ZnT2 causes a defect in zinc secretion, which leads to a reduced zinc content in the breast milk during lactation. Consequently, infants that are exclusively breast-fed develop a zinc deficiency, which can be alleviated by oral zinc supplementation [67].

Zinc homeostasis is furthermore controlled by metallothioneins (MT), which are cysteine-rich 6–7 kDa proteins that bind metal ions such as zinc [68]. So far, MTs 1–4 are known in humans to play a coordinated role in the transport, distribution, and maintenance of intracellular zinc levels [68,69]. Up to 20% of intracellular zinc is bound to MTs and can be rapidly released. MTs are known to modulate three fundamental processes: (1) release of gaseous mediators (e.g., nitric oxide); (2) apoptosis, and (3) binding/exchange of heavy metals such as zinc, cadmium, or copper. Their main functions include, amongst others, detoxification of heavy metals such as mercury and cadmium; homeostasis of essential metals, including copper and zinc; antioxidation against reactive oxygen species (ROS); protection against DNA damage; and modulating proliferation and apoptosis. Due to their ROS-scavenging properties, they help to protect against several types of environmental stress. MT-1 and MT-2 are ubiquitously expressed throughout the human body, especially in the kidney, liver, intestine, and pancreas, and are involved in heavy metal detoxification, immune function, and in a variety of G.I. tract functions [69,70,71]. MT-3 and MT-4 are expressed more specifically: MT-3 is mainly present in the nervous system, but also in heart, retina, kidney, breast, prostate, bladder, and reproductive organs, and is involved in the development, organization, and apoptosis of brain cells. MT-4 is found predominantly in the stratified squamous epithelium, e.g., the upper G.I. tract, where it helps to regulate stomach acid pH, taste, and texture discrimination of the tongue [71]. MT-1 and MT-2 are inducible by various stimuli such as metal ions, proinflammatory cytokines, such as interleukin (IL)-1, IL-6, interferon (IFN)-α, tumor necrosis factor (TNF)-α, glucocorticoids, and oxidative stress, whereas the expression of both MT-3 and MT-4 is very strictly controlled [69,70,72,73].

Besides intracellularly, MTs have been also found to be released in a very small amount to the extracellular environment, like it has been reported for serum, urine, bronchoalveolar spaces, liver sinusoids, and inflammatory lesions. Although MTs are present at very low levels in the extracellular environment, they support the beneficial movement of leukocytes to the site of inflammation by providing a “danger signal” to the cells and alter the character of the immune response when cells perceive cellular stress [74]. Regarding inflammation, MTs play a critical role in activating immune cells by increasing the intracellular concentration of free zinc ions and modulating proinflammatory cytokines on the one hand [55]. On the other hand, MTs themselves are synthesized in response to proinflammatory acute phase cytokines released at sites of inflammation. Glucocorticoids, a signal often associated with a stressful environment, also trigger MT expression. In addition, MTs can be induced by ROS, endotoxins, and divalent metal cations, such as zinc [74]. With all these different triggers, it is not surprising that increased MT levels are found in association with neoplastic diseases [75], autoimmune diseases [76], chronic inflammation [77], and infections [78,79]. Furthermore, it is described that MTs have immunomodulatory activities, such as reducing T-cell-dependent humoral responses, altering the proliferative capacity of lymphocytes, and decreasing cytotoxic T-cell function and macrophage effector function.

The various expression patterns and different functions illustrate the importance of MT isoforms for distinct organ and tissue function and clearly suggest a strong interplay between MTs and the immune system. Hence, insufficient expression of MTs in the context of inflammatory diseases can dramatically shorten lifespan [74].

In addition to MTs, members of the S100 protein family can also bind intracellular zinc, hence regulating zinc homeostasis [80]. All 20 S100 family proteins act in a calcium-dependent manner and have calcium-buffering capacities, whereas some proteins are also regulated by zinc and several members show higher affinity towards zinc compared to calcium, such as S100A3 [81]. Zinc-binding S100 proteins include S100B, S100A1, S100A2, S100A3, S100A5, S100A7, S100A8/9, S100A12, and S100A16. Of these, calprotectin and calgranulin (S100A8/9 and S100A12) play a crucial role in nutritional immunity, as mentioned later [82]. S100 family proteins are a broad subfamily of low-molecular-weight calcium-binding proteins (9–14 kDa) with structural similarity distributed in a cell-specific, tissue-specific, and cell-cycle-specific manner, and having functional discrepancy. Their multitude of functions range from calcium buffering, modulating enzyme activity and enzyme secretions, modulating apoptosis and transcription, proliferation and differentiation, cellular chemotaxis, and much more [82,83]. The family of S100 proteins is a critical connecting link in innate immunity. It facilitates the immune response cascade through direct participation, and provides host defense mechanisms by triggering immunological responses against invading pathogens.

Zinc sequestration is furthermore facilitated by α2-macroglobulin (A2M), which is an inhibitor of matrix metalloproteases (MMPs). A2M has a very high affinity to zinc, and zinc is needed for A2M activation and binding to cytokines [58]. A2M expression is triggered due to the expression of proinflammatory acute-phase proteins, including IL-6. Hence, zinc-binding peptides such as MTs and A2M are upregulated during inflammation, facilitating a reduction in zinc availability for invading pathogens [84]. This mechanism is beneficial during acute immune response; however, a long-term decrease in zinc availability may contribute to pathological processes in conditions of chronic inflammation, like it is seen in diabetes or dementia [11,12,13]. Moreover, high IL-6 expression and consequently upregulated MT and A2M expression is linked to immunosenescence and increased risk of Alzheimer’s disease [58].

## 5. Zinc and Nutritional Immunity

Antimicrobial peptides and/or proteins (AMPs) play an essential role in the first line of defense against a wide range of pathogens [85]. In humans, many AMPs, bactericidal factors, and host defense peptides exist [82,85]. For instance, calcium-dependent proteins S100A7 (psoriasin), S100A15, the S100A8/A9 heterodimer (calprotectin), and calgranulins (calprotectin and S100A12) belong to the AMPs [82].

Calgranulins, in particular, take advantage of this mesmerizing property, and inhibit microorganism growth by essential-nutrient deprivation, thereby limiting pathogen invasion and growth [86,87].

Innate immune cells, such as neutrophil granulocytes, are the first line of cell-based immunological defense and are well known for calprotectin expression. Calprotectin is the most abundant protein in neutrophil granulocytes and function as antimicrobial protein via zinc-chelating capacity, which causes zinc depletion through sequestration. Hence, pathogens such as *Candida albicans*, *Acinetobacter baumannii*, *Klebsiella pneumoniae*, *Helicobacter pylori*, *Escherichia coli*, and *Staphylococcus aureus* are in poor condition, leading to so-called nutritional immunity [82,88]. Additionally, neutrophils are able to release neutrophil extracellular traps (NET) that consist of cellular DNA, chromatin, and granular proteins. NETs are a mechanism to capture and finally kill microorganisms, which is accompanied by simultaneous calprotectin release in a high concentration to support the elimination of the invading pathogens [89]. This mechanism is well known for host defense against bacteria, and was also recently shown for viruses, such as COVID-19 [90], and fungi, as shown for the world’s three most deadly fungal pathogens (*C. albicans*, *Aspergillus fumigatus*, and *Cryptococcus neoformans*) [91]. According to the National Nosocomial Infections Surveillance System study, *Candida* species are responsible for 85% of fungal infections among patients in critical conditions. Candidemia commonly results in a high rate of morbidity and mortality, as well as high medical expense, among patients who are hospitalized. Hence, fungal control plays a decisive role in human health [92].

Interestingly, immune cells of either innate and adaptive immunity have developed two opposing strategies to eliminate invading pathogens:

On the one hand, they are able to reduce the zinc content of the phagosome or the cytoplasm to limit zinc availability, like it has been described for macrophages in *Histoplasma capsulatum* infection, resulting in nutritional immunity [93]. Additionally, zinc transporter ZIP8 is described to enable the same effect in activated T cells [94]. On the other hand, the zinc content of the phagosome is excessively increased, leading to zinc intoxication of the pathogen, as known, for instance, for *Mycobacterium tuberculosis* infection [95]. Hence, nutritional immunity and metal intoxication are feasible immune strategies to limit pathogen growth and to eventually control infection. Nutritional immunity mainly affects enzymatic and metabolic function, while metal overload contributes to reactive oxygen species and reactive nitrogen species, protein mismetallization, and subsequent respiratory arrest. Both are well described for several pathogens, including bacteria, viruses, and fungi [88,96,97,98].

In general, during infection, the overall plasma zinc level is reduced to limit zinc availability for pathogens. This is primarily facilitated by upregulation of the zinc transporter ZIP 14, which is induced in an IL-6-dependent manner. This occurs rapidly within only a few hours and leads to zinc accumulation in the liver and hypozincemia in the serum, respectively [99,100].

Although different strategies are known to limit zinc environment for the invading pathogen, some pathogens have developed defense strategies to overcome some mechanisms [96,101]:

Neisseria—*N. gonorrhoeae* and *N. meningitidis*—have evolved mechanisms to acquire transition metals such as manganese, iron, copper, and zinc in specific environments. Gonococcal TdfH and TdfJ are zinc-specific transporters that pirate zinc from calprotectin and psoriasin, respectively, to transport zinc across the outer cell membrane into the periplasmatic space. *N. meningitides* expresses a similar zinc transporter called CbpA. Both *N. gonorrhoeae* and *N. meningitides* additionally express a specific zinc-import system called ZnuC, ZnuB, and ZnuA to improve intracellular zinc status. Moreover, *N. meningitidis* uses ZnuD to escape NET-mediated nutritional immunity [102]. Znu-like systems are found in a wide variety of Gram-negative and Gram-positive bacteria [103], such as *E. coli*, *M. tuberculosis*, *Bacillus subtilis*, *Salmonella enterica* and S. *typhi*, *K. pneumoniae*, *Yersinia pestis*, and much more [103,104,105,106]. Another category of zinc transporters is similar to the eukaryotic ZIP family transporters; however, ZIP homologs have only been identified in *E. coli* [103]. In summary, a lot of pathogens have evolved mechanisms to circumvent nutritional immunity as zinc deficiency.

Zinc deficiency is harmful not only for pathogens, but also for humans themselves. Low zinc is associated with an impaired immune system and poor prognosis in various diseases as common cold, sepsis, malaria, and much more [1,24,107]. Zinc supplementation can be very supportive, as shown in a multitude of studies, investigating, for example, lower respiratory tract infection, diarrhea, pneumonia, or common cold. For the latter, zinc sulfate or zinc gluconate via lozenges at concentrations of about 75 mg/day reduced the mean daily clinical score, nasal secretion weight, and viral shedding, and made more symptom-free episodes possible [108,109,110,111,112,113,114]. Similar results are known for pneumonia in children. Zinc supplementation of 20 mg/day enhanced recovery and reduced resistance to antimicrobials through decreasing exposure to broad-spectrum antibiotics [115]. For diarrhea, zinc supplementation was shown to reduce the incidence and prevalence [16]. This has been extensively studied in numerous investigations, and the results show that zinc reduces the overall duration, severity duration, severity, and frequency of stools in children and adults. Moreover, consequential damage of the intestine is also reduced [116]. This is of main importance, since diarrheal infections count for the second leading cause of death in children under five years of age in developing countries. Hence, the WHO and United Nation’s Children Fund (UNICEF) recommend oral rehydration solutions and 10–14 days’ supplemental zinc treatment in acute diarrhea, particularly in children [117].

However, supplementation during some diseases seem to be not beneficial, as shown for sepsis [118], and some contradictory results also exist showing no beneficial effect of zinc supplementation for different diseases, although this has been presented before (see Table 1 and Table 2) [119,120,121,122,123].

It is important to consider the risk of creating a zinc microenvironment that favors pathogen growth when zinc is supplemented during infection while interfering with the efforts of the innate system to chelate available free zinc. The importance of understanding the complex interplay between different nutrients and their collective influence on health in more detail has been prominent for decades.

On the one hand, a deficiency of one micronutrient is often accompanied by a deficiency of another micronutrient, or even by a general micronutrient deficiency, as occurs particularly in the elderly or in people on a plant-based diet, often due to malnutrition [14,48,124]. On the other hand, high-dose supplementation or overdosing of one micronutrient is critical, since several micronutrients influence each other, and thus an imbalance of another micronutrient may occur. For example, iron supplementation is known to interfere with the proper absorption of zinc, resulting in zinc deficiency.

Moreover, zinc supplementation in high doses compromises copper uptake and vice versa. Adequate zinc and copper homeostasis is known to be important for decades and an altered copper/zinc ratio is found in multiple diseases, such as neurodegenerative disorders [125,126], hepatocellular and gastric carcinoma [127,128], immunological disorders such as sickle cell disease [129], and bacterial and viral infectious diseases such as leishmaniosis or COVID-19 [130,131]. In future, it may possible that the copper/zinc ratio may be considered as a biomarker for disease or for mortality in the elderly population. In advance, more research needs to be performed to understand the therapeutic potentials of micronutrient supplementation, particularly zinc supplementation, on numerous diseases.

In recent years, several clinical trials have been conducted that investigated zinc supplementation in various diseases. Studies are sorted by type and/or origin of infectious disease into viruses (Table 1), bacteria (Table 2), and parasites/fungi (Table 3). The influence of zinc depends on many factors, including the following: zinc status at baseline measurement, zinc formula administered, zinc supplementation concentration, zinc supplementation frequency and type of supplementation (preventive or therapeutic), and subject’s age.

**Table 1 biomolecules-12-01748-t001:** Zinc supplementation and viral diseases.

Disease	Zinc Salt/Formulation	Period	Population	Effect of Supplementation	References
**Common cold**	Zinc gluconate, 23.0 mg 6 times/d	7 days	C:28 Z: 37	Shortened duration of cold, more zinc-treated subjects are asymptomatic compared to control subjects	[108]
	Zinc gluconate, 23.0 mg 6 times/d	6 days	P: 28 Z: 29	Reduced mean daily clinical score and nasal secretion weight and viral shedding	[114]
	Zinc gluconate, 23.0 mg 8 times/d	5 days	P: 16 Z: 16	No significant difference between nasal symptom scores, same median duration of viral shedding	[132]
	Zinc gluconate, 23.0 mg 6 times/d	10 days	75	Shortens duration and reduces symptom severity	[109]
	Zinc gluconate, 13.3 mg 6 times/d	8 days	P: 50 Z: 50	Shortens duration and reduces symptom severity, especially cough, headache, nasal congestion/drainage	[133]
	Zinc gluconate, 4.5 mg 4–6 times/d	10 days	P: 69 Z: 61	No benefit was observed among the groups	[119]
	Zinc acetate, 10.0 mg 4 times/d	6 days	P: 28 Z: 30	No benefit was observed among the groups	[120]
	Zinc acetate, 9.0 mg 6 times/d	14 days	P: 49 Z:52	Overall symptom duration was significantly less	[110]
	Zinc acetate, 12.8 mg 6 times/d	12 days	P: 23 Z: 25	Reduced duration and severity of cold symptoms, especially cough	[111]
	Zinc gluconate, 13.5 mg Zinc acetate, 11.5 mg or 5.0 mg 6 times/d	14 days	P: 67 ZG: 69 ZA_5.0_: 66 ZA_11.5_: 70	Zinc gluconate treatment: reduced median duration of symptoms Zinc acetate lozenges: no effect on the duration or severity of symptoms	[112]
	Zinc sulfate, 15.0 mg/d	7 month	P: 100 Z: 100	Mean number of colds in the zinc group was significantly fewer	[134]
	Zinc gluconate, 15.0 mg/d	7 month	P: 17 Z: 17	More symptom-free episodes	[113]
**HIV/AIDS**	Zinc sulfate, 45.5 mg/d	1 month	P: 29 Z: 29	Increase or stabilization in body weight, increase in plasma zinc levels, CD4^+^ T cells and plasma active zinc-bound thymulin; reduced or delayed frequency of opportunistic infections due to *Pneumocystis jirovecii* and *C. albicans*	[135]
	Zinc gluconate, 45.0 mg 3 times/d	15 days	P: 5 Z: 5	Increased zinc concentrations in red blood cells, HLA-DR^+^ cells, stimulation of lymphocyte transformation, and phagocytosis of opsonized zymosan by neutrophils	[136]
	Zinc sulfate, 10.0 mg elemental zinc/d	6 month	P: 41 Z: 44	Decreased morbidity from diarrhea	[137]
	Zinc sulfate, 50.0 mg/d	1 month	P: 34 Z: 31	No improvements in immune responses to tuberculosis, CD4/CD8 ratio, lymphocyte subsets, and viral load	[138]
	Zinc sulfate, 25.0 mg/d	6 month	P: 200 Z: 200	No effect on birth outcomes by supplementation to pregnant HIV-positive women, no effect on T-lymphocyte counts	[139]
	Zinc sulfate, 25.0 mg/d	6 month	P: 200 Z: 200	Increased risk of wasting, no effect on viral load	[140]
	Zinc gluconate, 50.0 mg/d	6 days	P: 45 Z: 44	No improvements in antibody responses to a pneumococcal conjugate vaccine	[141]
	12.0–15.0 mg zinc/d (women–men)	18 month	P: 116 Z: 115	Four-fold reduction in the likelihood of immunological failure, reduced rate of diarrhea	[142]
	Chelated zinc, 15.0 mg/d	12 month	P: 17 Z: 13	CD4^+^ cell count significantly increased	[143]
	Daily zinc intake (not specified)	18 month	P: 128 Z: 126	Nonsignificant decrease in Veterans Aging Cohort Study (VACS) index	[144]
	Zinc sulfate, 15.0 mg/d	12 month	P: 40 Z:40	No benefit was observed among the groups	[145]
	Zinc gluconate, 12.0–15.0 mg/d (women–men)	18 month	P: 128 Z: 126	No benefit was observed among the groups	[146]
	Zinc gluconate (high (Z_hi_) zinc): 90.0 mg elemental/d, Zinc gluconate (low (Z_low_) zinc): 45.0 mg elemental/d	16 weeks	Z_hi_: 27 Z_low_: 25	Increased serum zinc, decreased biomarkers associated with clinical comorbidities (decreased systemic inflammation (c reactive protein and TNF-α), monocyte activation, and enterocyte damage)	[147]
	Zinc sulfate, 20.0 mg/d	24 weeks	P: 26 Z: 26	Decrease in viral load, anthropometric indices, and morbidity profile in HIV-infected children started on antiretroviral therapy	[148]
**Chronic HepC**	Polaprezinc, 75.0 mg 2 times/d	24 weeks	P: 35 Z: 40	Improved response to IFN-α therapy	[149]
	Zinc sulfate, 300.0 mg/d Polaprezinc, 150.0 mg/d	24 weeks	P: 10 Z_S_: 9 Z_P_: 15	Normalized serum ALT levels, better eradication of HepC virus RNA	[150]
	Zinc gluconate, 78.0 mg/d 5 times/d	6 month	P: 40 Z: 18	Decreased incidences of gastrointestinal disturbances, body weight loss, and mild anemia	[151]
	Zinc gluconate, 30.0 mg/d	1 year	C: 16 Z: 16	No benefit was observed among the groups	[122]
	Polaprezinc, 75.0 mg 2 times/d	48 weeks	P: 16 Z: 16	No benefit was observed among the groups	[152]
	Polaprezinc, 75.0 mg 2 times/d	24 weeks	P: 39 Z: 39	No benefit was observed among the groups	[153]
	Polaprezinc, 75.0 mg 3 times/d	6 month	P: 12 Z: 12	Reduced serum AST, ALT, and ferritin	[154]
	Polaprezinc, 150.0 mg 2 times/d	6 years	P: 30 Z: 32	Reduced incidence of HCC (albumin-dependent)	[155]
	Polaprezinc, 150.0 mg/d	48 weeks	P: 12 Z: 11	Decrease in serum ALT levels and Th2 cells (%), decreased plasma thiobarbituric acid reactive substances, and prevented decrease in polyunsaturated fatty acids of erythrocyte membrane phospholipids	[156]
**Dengue fever**	Zinc bisglycinate, 15.0 mg 3 times/d	5 days	P: 25 Z: 25	Lower mean time of defervescence and shorter time of hospitalization	[157]
**COVID-19**	Zinc sulfate, 100.0 mg elemental/d	Until recovery or death	P: 46 Z: 196	No benefit was observed among the groups	[123]
	Zinc sulfate, 100.0 mg elemental zinc/d	5 days	P: 521 Z: 411	Increased frequency of being discharged home (OR 1.53, 95% CI 1.12–2.09) and reduction in mortality or transfer to hospice among patients who did not require ICU level of care	[158]
	Zinc sulfate, 100.0 mg elemental/d	4 days	P: 2467 Z: 1006	Increased rates of discharge home and 24% reduced risk of in-hospital mortality	[159]
	Zinc sulfate, 50.0 mg elemental/d	5 days	P: 377 Z: 141	Fewer hospitalizations	[160]
	Zinc acetate or gluconate, 2.0–2.5 mg/kg/day	10 days	Case report Z: 28	Short recovery time (~1,6 days)	[161]
	Zinc chloride, 0.24 mg/kg/d i.v.	7 days	P: 18 Z: 15	Increased zinc level but no effect on clinical outcome	[162]
	Zinc sulfate, 50.0 mg elemental/d 2 times/d	15 days	P: 95 Z: 96	Decreased duration of ventilation, decreased length of hospitalization, and reduced risk of in-hospital mortality	[163]
	Zinc bisglycinate, 15.0 mg/d	6 weeks	P: 57 Z: 59 C: 56	Significant decrease in SARS-CoV-2 infection	[164]

P: placebo, Z: zinc, C: control, ZA: sinc acetate, ZG: zinc gluconate.

**Table 2 biomolecules-12-01748-t002:** Zinc supplementation and viral diseases.

Disease	Zinc Salt/Formulation	Period	Population	Effect of Supplementation	References
**Acute lower respiratory tract infection**	Zinc bisglycinate, 30.0 mg elemental/d	7 days	P: 32 Z: 32	Shortened recovery time and duration of the hospital stay, and improved chest in-drawing, tachypnea, and fever	[157]
	Zinc sulfate, 20.0 mg/d	5 months	P: 124 Z: 134	Reduced acute lower respiratory tract infection morbidity	[165]
	Zinc gluconate, 10.0 mg/d	60 days	P: 48 Z: 48	Reduced episodes of acute lower respiratory infections and severe acute lower respiratory infections, increased infection-free days	[166]
	Zinc oxide, 5.0 mg/d	12 months	P: 167 Z: 162	Decreased incidence of upper respiratory tract infections and diarrheal disease episodes	[167]
	Zinc gluconate, 10.0 mg/d	6 months	P: 311 Z: 298	Increased plasma zinc level and decreased episodes of infection	[168]
	Zinc acetate, 10.0 mg 2 times/d	5 days	P: 74 Z: 76	Increased recovery rates from illness and fever in boys	[169]
	Zinc sulfate, 15.0 mg/d	6 months	P: 40 Z: 40	Increased plasma retinol concentrations, earlier sputum conversion and resolution of X-ray lesion area	[170]
**Pneumonia**	10.0–20.0 mg zinc/d	2 weeks	P: 280 Z: 280	Acceleration in clinical resolution and shorter hospital stay	[171]
	10.0–20.0 mg zinc /d	2 weeks	610	Marginal faster recovery time	[172]
	Zinc sulfate, 10.0 mg 2 times/d	until discharge	299	No benefit was observed among the groups	[173]
	Zinc sulfate, 12.5 mg elemental/d 2 times/d	until discharge	P: 47 Z: 47	No benefit was observed among the groups	[174]
	Elemental zinc, 20.0 mg/d	until discharge	P: 84 Z: 80	Reduced duration of severe pneumonia, duration of chest in-drawing respiratory rate hypoxia, and overall hospital duration	[115]
	Zinc sulfate, 10.0–20.0 mg/d	7 days	P: 301 Z: 303	Faster recovery from lower chest wall indrawing and sternal retraction	[175]
	Zinc syrup, 20.0 mg elemental/d 2 times/d	until discharge	P: 225 Z: 225	Faster resolution of respiratory signs	[176]
	Elemental zinc, 10.0 mg 2 times/d	7 days	P: 53 Z: 64	No benefit was observed among the groups	[177]
	Zinc gluconate, 10.0 mg/d–20.0 mg/d	7 days	P: 176 Z: 176	No benefit was observed among the groups	[178]
	Zinc syrup, 40.0 mg/d	until discharge	P: 150 Z: 150	Shorter duration of relief of severe pneumonia signs and hospitalization time	[179]
	Zinc syrup, 10.0 mL/d	until discharge	P: 60 Z: 60	Faster resolution of clinical symptoms	[180]
**Tuberculosis**	Zinc sulfate, 220.0 mg/d	18 months	Z: 8	Reduced dose of clofazimine, withdrawal of steroids, toleration of dapsone, reduced incidence and severity of erythema nodosum leprosum, gradual decrease in the size of granuloma, and gradual increase in the number of lymphocytes	[181]
	Zinc sulfate, 15.0 mg/d, +/−VitA 5000 IU/d	6 months	P: 40 Z: 40 VitA: 40 Z+VitA: 40	Marginal earlier sputum conversion No difference in clinical, nutritional, chest X-ray, or laboratory findings	[182]
	Zinc sulfate, 30.0 mg elemental every second day	6 month	P: 37 Z: 37	Elevated plasma zinc concentrations, elevated body weight, earlier sputum smear conversion, lower SGOT and SGPT concentrations after 2 months, decreased serum levels of total protein and albumin	[183]
	Zinc sulfate, 15.0 mg/d	6 month	P: 40 Z: 40	Increased plasma retinol concentrations, earlier sputum conversion and resolution of X-ray lesion area	[170]
**Shigellosis **	Zinc acetate, 1.30 mg/kg 3 times/d	1 month	P: 16 Z: 16	Increased intestinal mucosal permeability and better nitrogen absorption, increased serum zinc and alkaline phosphatase activity	[184]
	Zinc acetate, 20.0 mg/d	2 weeks	P: 28 Z: 28	Increased serum zinc level, lymphocyte proliferation in response to phytohemagglutinin and plasma invasion plasmid-encoded antigen-specific IgG titers	[185]
	Zinc acetate, 20.0 mg/d	2 weeks	P: 28 Z: 28	Increased serum zinc levels, serum shigellacidal antibody titers, CD20^+^ cells, and CD20^+^CD38^+^ cells	[186]
	Not specified, 20.0 mg/d	2 weeks	P: 16 Z: 14	Faster recovery from acute illness, increased mean body weight, and fewer episodes of diarrhea	[187]
* **H. pylori** *	Polaprezinc, 150.0 mg 2 times/d	7 days	P: 28 Z: 33	Increased cure rate of *H. pylori* infection compared to single antibiotic treatment	[188]
**Diarrhea**	Zinc sulfate, 3.0–7.0 mg/kg/d elemental zinc/d	4 month	P: 70 Z: 70	Decreased incidence of diarrhea, number of diarrhea episodes per child, and frequency of stools per day	[189]
	Diarrhea multiple different studies			Decreased duration, severity, and occurrence of diarrhea	[190]
	Zinc acetate, zinc gluconate, zinc sulfate ranging from 5.0–40.0 mg/d	5–15 days	P: 9353 Z: 9469	Reductions in morbidity as a result of oral therapeutic zinc supplementation for acute diarrhea among children	[191]
	Zinc sulfate, 10.0 mg/d	2 weeks	P: 536 Z: 538	No benefit was observed among the groups	[121]
	10.0–20.0 mg zinc/d	until discharge	P: 50 Z: 50	Reduced frequency of diarrheal episodes	[192]
	Zinc gluconate syrup, 20.0 mg elemental/d, +/− daily Probiotics (Pr)	7 days	P: 50 Pr: 50 Z: 46	Reduced relative risk of diarrhea persistence, decreased duration and severity, reduced post-treatment complications	[193]
	Zinc tablet, 7.0 mg/d, therapeutic zinc (TZ), 20.0 mg/d, +/− micronutrient powder (MNP)	9 month	P: 847 Z: 844 TZ: 848 MNP: 841	No benefit was observed among the groups	[194]
	Zinc sulfate, 10.0–20.0 mg/d	10 days	P: 50 Z: 53	Reduced duration of diarrhea, fewer diarrheic episodes in the next 3 months	[195]

P: placebo, Z: zinc, C: control, VitA: vitamin A, Pr: probiotics, MNP: micronutrient powder, TZ: therapeutic zinc.

**Table 3 biomolecules-12-01748-t003:** Zinc supplementation and parasitic or fungal diseases.

Disease	Zinc Salt/Formulation	Period	Population	Effect of Supplementation	References
**Malaria**	10.0 mg zinc 6 times/week +/− VitA single dose, 200,000 IU	6 months	P: 74 Z: 74	Decreased malaria prevalence and fewer malaria episodes, longer time to first malaria episode, and 22% fewer fever episodes	[196]
	Zinc gluconate, 10.0 mg 6 times/week	46 weeks	P: 138 Z: 136	Reduction in Plasmodium falciparum-mediated febrile episodes	[197]
	Zinc acetate or zinc gluconate, 70.0 mg 2 times/week	15 month	P: 54 Z: 55	Not statistically significant trend towards fewer malaria episodes; no effect on plasma and hair zinc, diarrhea, and respiratory illness	[198]
	Zinc sulfate, 12.5 mg 6 times/week	6 months	P: 344 Z: 336	Increased serum zinc levels and reduced prevalence of diarrhea	[199]
	Zinc sulfate, 20.0 mg or 40.0 mg/d	4 days	P: 483 Z: 473	Increased plasma zinc, no effect on fever, parasitemia, or hemoglobin concentration	[200]
	Zinc sulfate, 20.0 mg/d	7 months	P: 189 Z: 191	No significant effect on *P. vivax* incidence but significantly reduced diarrhea morbidity	[201]
	Zinc sulfate, 25.0 mg/d +/− VitA 2500 IU/d	until delivery	P: 362 VitA: 348 Z: 345 VitA+Z: 349	36% (95% CI = 9–56%) reduced risk of histopathology-positive placental infection	[202]
	5.0 mg, 10.0 mg, or 15.0 mg zinc/d	9 month	P: 785 C: 433 Z5: 429 Z10: 438 Z15: 436	No benefit was observed among the groups	[203]
	Zinc gluconate, 10.0 mg/d +/− VitA, 200,000 IU/d at the beginning and end of the study	6 month	C: 90 Z: 92	Significantly fewer (27%) malaria diagnoses	[204]
	Zinc, 10.0 mg/d +/− Multi-nutrients (M)		P: 148 M: 148 Z: 145 M + Z: 146	No benefit was observed among the groups	[205]
**Leishmania infection**	Zinc sulfate, 2.5 mg/kg, 5.0 mg/kg or 10.0 mg/kg 3 times/d	45 days	P: 12 Z: 92	Increased serum zinc levels and cure rate, decreased erythema and size of induration	[206]
	Zinc, 45.0 mg/d	20 days	P: 15 Z: 14	Higher expression level of transferrin receptor	[207]
	Zinc sulfate, topical 2%	3 month	P: 32 Z: 32	No benefit was observed among the groups	[208]
	Zins syrup, total dose in 2 weeks of 2 mg/kg/d	2 weeks	C: 26 Z: 26	Accelerated reduction in splenomegaly	[209]
	Zinc sulfate, 10.0 mg/kg/d	45 days	C: 50 Z: 50	Zinc supplementation is as effective as systemic meglumine antimoniate treatment	[210]
	Zinc sulfate, itralesional injections of 2% zinc solution	6 weeks	C: 35 Z: 31	Higher efficacy after the second and fourth weeks	[211]
* **C. albicans** *	Zinc syrup, 20.0 mg elemental/d	2 weeks	P: 366 Z: 358	Increased blood zinc concentration, reduced prevalence of candidemia and candiduria by 50%, lesser nosocomial urinary tract infection and bloodstream infection, shorter treatment with broad-spectrum antibiotics, shorter length of hospital stay	[212]

P: placebo, Z: zinc, C: control, VitA: vitamin A, M: multinutrients.

## 6. Zinc and the Inflamed Immune System

Invading pathogens put the healthy human body into a state of inflammation by alerting and activating the immune system to fight those pathogens and to eventually restore the healthy state. Inflammation is a natural and necessary process that is required to protect the host from tissue damage; however, sometimes overshooting immune reactions occur that cause severe tissue damage or inflammation processes, which are not resolved and later become chronic, leading to loss of tissue function [213]. Nonresolving inflammation contributes to many diseases, including COVID-19 in its fatal and long forms, sepsis, or multiple organ failure, which are a major cause of admission to the intensive care unit and high mortality rates [214].

Since 2010, the toll that inflammation takes on human health has not abated, despite major advances in understanding the underlying biology. Nevertheless, a lot of research has been performed regarding drug development, such as biologics that block signaling through IL-1β and TNF-α, or regarding nutritional supplements, including zinc, that improve the overall immune status of patients suffering of severe diseases [13,214,215].

In immunological pathogen defense, neutrophil leukocytes (polymorphonuclear leukocytes (PMNs)) and macrophages are among the first responders to infection or tissue damage [216]. The first signals responsible for the inflammatory processes are initially activated by damage-associated molecular patterns (DAMPs), such as DNA and extracellular components or ATP, at the tissue injury site. After cellular activation through pattern recognition receptors (PRR), immune cells migrate to the site of injury, which is called chemotaxis. PRR can be subdivided into three groups: toll-like receptors, C-type lectin receptors, and retinoic acid-inducible gene 1-like receptors [217]. The early recruitment of PMN can be mediated by a complex interplay of chemokines, proinflammatory cytokines, complement factors, and bacterial cell wall compounds, which is highly affected by the patient’s zinc status. It is reported that during zinc deficiency, chemotaxis is reduced, and can be restored by physiological zinc supplementation [218,219,220]. Early activation of PMN functions, amongst others, through the SCR family kinase LYN, which is known to be regulated in a zinc-dependent manner. In all leukocytes, the SCR family kinase is prevalent. Activation and inhibition are mediated on the molecular level by phosphorylation, which is facilitated by kinases such as protein tyrosine phosphatases (PTP). For example, CD45 is a transmembrane PTP that is an important regulator of SCR family kinase activity. Since the inhibition constants of zinc for PTPs are very low (varying from micromolar to picomolar concentrations [221]), cellular zinc levels are sufficient to alter enzymatic activity of PTP that eventually regulate cellular activation and function [222].

Besides chemotaxis, phagocytosis and pathogen killing are controlled by zinc status. Following phagocytosis, pathogens are killed by the activity of nicotinamide adenine dinucleotide phosphate (NADPH) oxidases that have been shown to be inhibited by zinc deficiency and zinc excess, respectively [223,224]. Alternatively, PMNs kill pathogens by releasing neutrophil extracellular traps (NETs), which are abolished during zinc deficiency [218]. NETs are web-like structures composed of DNA, chromatin, fibers, and various neutrophil granule proteins [225]. NETosis is a double-edged sword for the immune system. On the one hand, it is known that NETs can encapsulate, capture, and kill pathogens efficiently, especially large microorganisms such as *C. albicans* and *Mycobacteria* aggregates that are difficult to phagocytose. On the other hand, the excessive release of NETs can trigger and amplify inflammatory diseases such as sepsis, rheumatoid arthritis, or inflammatory bowel disease [225]. However, NETosis is an indispensable method to protect humans of invading harmful pathogens.

Due to pathogens, monocytes circulating in the blood may be activated and migrate to the damaged tissue. Therefore, they need to adhere to endothelial cells, which can be intensified by zinc supplementation. During zinc deficiency, this mechanism is impaired, as is, for example, well reported for the elderly population suffering of zinc deficiency very often [226]. Moreover, proinflammatory cytokine production, such as IL-1, IL-6, and TNF-α, is increased. In line with that, studies uncovered a rapid decline of plasma zinc concentrations during the acute phase of infection, leading to hypozincemia [227,228,229]. Hence, redistribution of zinc during inflammation seems to be mediated by proinflammatory cytokine production. Zinc is shuttled into cellular compartments to assist protein synthesis, neutralization of free radicals, and to prevent microbial survival.

One of the most important mediators in inflammation is the nuclear factor kappa-light-chain-enhancer of activated B cells (NFκB) (Figure 1). Besides inflammatory process regulation, it controls apoptosis, cell adhesion, proliferation, tissue remodeling, the innate and adaptive immune responses, and cellular stress responses. NFκB activation requires the phosphorylation of IKKα/β by the IκB kinase (IKK) complex, which degrades the IκB and releases NFκB and allows it to translocate freely to the nucleus to induce targeted gene expression [230,231].

Over the last few decades, contradictory studies have reported zinc to have activating and inhibiting properties on this process, respectively. This is a great and complex example to illustrate how differently zinc is able to fine-tune cell signaling depending on which model is considered. Thus, it is obvious that the influence of zinc cannot be interpreted using a unilateral approach [232].

In vitro studies revealed that increased intracellular zinc concentrations are able to inhibit dephosphorylation of mitogen-activated protein kinase (MAPK) and to support phosphorylation of IKKα/β. Hence, NFκB can translocate into the nucleus and activate expression of proinflammatory cytokines [233]. Using chelating agents to prevent zinc signals, such as TPEN (N,N,N0,N0-Tetrakis(2-pyridylmethyl)ethylenediamine), NFκB is not able to translocate to the nucleus anymore. On the one hand, this might be due to the inhibition of kinases ERK1/2, IKKβ, MKK3/6 and IκB, which might be associated with the effect of zinc on PTP and PTK activities as described earlier [234,235]. On the other hand, IRAK1 destruction might be avoided, causing its accumulation in the cytosol. This might be due to the inhibition of zinc-dependent MMPs, which are probably responsible for IRAK1 destruction during adequate zinc conditions. IRAK1 is responsible for the degradation of NFκB inhibitors by IKK.

In contrast, a growing body of literature supports the role of zinc as a negative regulator of NFκB signaling:

For instance, zinc supplementation is described to influence the zinc-finger protein A20 expression, which is an anti-inflammatory protein. A20 negatively regulates tumor necrosis factor receptor (TNFR)- and toll-like receptor (TLR)-initiated NFκB pathways. Expression of A20 mRNA and protein expression can be induced by zinc, as shown in premonocytic, endothelial, and cancer cells [236]. A20 deubiquitinates receptor-interacting protein 1 (RIP1), which prevents its interaction with NFκB essential modulator IKK. Additionally, it inhibits TLR signaling by removing polyubiquitin chains from TNFR-associated factor 6 (TRAF6). Moreover, zinc supplementation was able to downregulate pro-inflammatory cytokine production by decreasing the gene expression of IL-1β and TNF-α through upregulation of mRNA- and DNA-specific binding for A20 [237]. In addition to zinc, proinflammatory cytokines are known inducers for A20 in macrophages, possibly representing a feedback mechanism to prevent hyperinflammatory reactions [238].

Proinflammatory pathways: In TNF-α and IL-1 signaling pathways, zinc supply prevents the translocation of nuclear translocation of NFκB and inhibits subsequent inflammation by modulation of the common IκB kinase complex that phosphorylates the NFκB inhibitory protein. Zinc also inhibits the IL-6-mediated activation of STAT3.

Anti-inflammatory pathways: In TGF-β signaling, zinc supply promotes Smad 2/3 nuclear translocation and transcriptional activity. Zinc promotes the IL-2 signaling pathway by blocking MAP kinase phosphatase (MKP) in extracellular signal-regulated kinase (ERK) 1/2 pathways and phosphatase and tensin homologue (PTEN), which opposes the phosphoinositide 3-kinase (PI3K) function in the PI3K/AKT pathway. In IL-4 signaling, zinc induces STAT6 phosphorylation, promotes dimerization and translocation into the nucleus, and thus promotes anti-inflammatory effects.

Elevated cGMP level also might be responsible for preventing NFκB activation and translocation into the nucleus. Zinc acts as an inhibitor of cyclic nucleotide phosphodiesterase (PDE). The cGMP level rises when PDE is inhibited, thus activating protein kinase A (PKA). PKA suppresses Raf-1 by phosphorylation and subsequently inactivates MAPK and NFκB signaling [239]. Similarly, studies on mast cell signaling show that zinc can bind to a zinc-finger-like motif found on protein kinase C (PKC), which inhibits PMA-mediated PKC translocation to the membrane. Thus, NFκB activity is indirectly inhibited [240].

Besides NFκB signaling, IL-6-mediated signaling is also very important for modulation of an adequate proinflammatory immune response. IL-6 is one of the most potent inflammatory cytokines, activating immune cells by binding to the IL-6 receptor alpha chain and tow gp130 molecules, which are consequently activated and function as signal transducers (Figure 1). As a result, MAPK pathways and Janus kinase (JAK)-signal transducer and activator of transcription 3 (STAT3) pathways are activated, leading to STAT3 activation and dimerization. In particular, the phosphorylation of STAT3 is broadly affected by zinc homeostasis. During zinc deficiency, the JAK-STAT3 signaling pathway is activated in an IL-6 dependent manner. IL-6 was found to be highly upregulated while zinc-deficient. These observations were confirmed by DNA methylation profile characterization, showing a progressive demethylation of IL-6 promoter correlating with IL-6 expression [241]. However, zinc reconstitution normalizes the effect, and zinc supplementation was shown to inhibit IL-6/IL-1β-induced STAT3 phosphorylation [135,242]. Moreover, STAT3 phosphorylation is known to be regulated by other kinases, such as the SH2 domain-containing phosphatases (Shp), which belong to a class of non-transmembrane protein tyrosine phosphatases [243]. Furthermore, alteration in the redox state, e.g., via oxidative stress, seems to alter STAT3 phosphorylation [244], illustrating the complexity of regulation. STAT3 seems to be an important mediator in cell proliferation and differentiation, especially during inflammatory diseases such as sepsis [244]. Studies highlight a fundamental role for zinc-dependent STAT3 regulation in the innate as well as in the adaptive immune response. Zinc chloride increased transient STAT3 phosphorylation in hematopoietic stem cells, maintaining pluripotency and cellular self-renewal. STAT3 inhibition abrogates the zinc-related effect, underlining the importance of STAT3 as a central molecule in hematopoiesis. Moreover, B cell proliferation was increased during zinc deficiency, mediated amongst others by increased STAT3 phosphorylation [245]. However, IL-4-induced STAT6 phosphorylation was diminished [245]. Since the IL-4-induced pathway is essential for the immunoglobulin (Ig) class switch to IgE, altered signaling could explain the increased susceptibility towards parasite infections during zinc deficiency. IL-6 is also important for the induction of terminal B cell development into plasma cells, providing an additional reason for the decrease in antibodies during zinc deficiency [246]. For T cells, zinc supplementation was shown to inhibit the IL-6-induced STAT3 signaling cascade that is essential for T helper cell (Th) 17 development [247]. In the elderly population, secretion of proinflammatory IL-6 is pathologically increased while T-cell activation is reduced, as is likewise the response to stimulation or vaccination [248]. Interestingly, all pathologies can be ameliorated by zinc intake, which highlights the importance of zinc for a balanced immune response.

IL-1 promotes cell proliferation but also induces the expression of inflammation-related genes such as IFN-γ. When IL-1 binds to the IL-1 receptor (IL-1R) an induction of an intracellular zinc signal is described. Following the general signal transduction pathway, the IL-1 receptor-associated kinase (IRAK) is activated, which causes NFκB translocation into the nucleus that alters gene activation (Figure 1). While zinc-deficient, the production of IL-1β has been reported to be elevated, as found in in vitro studies of PBMC of zinc-deficient adults [249]. In line with that, epigenetic changes due to long-term zinc deficiency are known to promote changes of the chromatin structures of IL-1 β and TNF-α promoters, enabling the expression of both genes [227]. In contrast, zinc supply was shown to decrease IRAK activity (similarly to the effect on TLR4-signaling), which results in repression of the memory Th17 response, hence favoring the anti-inflammatory immune response [52,250].

An in vivo rodent model showed that zinc supplementation skewed the regulatory T cell (Treg)-Th17 balance towards Treg cells. After a pulmonary fungal challenge with *H. capsulatum*, zinc impaired dendritic cells (DC) to mount a proinflammatory response to fungal infection and stimulation by TLR ligands. Hence, zinc triggers the tolerogenic phenotype of DC and promotes the T-cell differentiation towards Treg cells [251]. In this context, zinc is known to keep unwanted T-cell-mediated immune responses in check by suppressing apoptosis, proliferation of immune cells, and production of proinflammatory cytokines, as well as IL-2 and IL-10 [252,253]. One important trigger for Treg development is the TGF-β-induced Smad 2/3 signaling pathway, which induces the expression of transcription factor forkhead-box-protein p3 (FoxP3). This is an important transcription factor for T-cell lineage development. Zinc supplementation amplifies TGF-β-induced Smad 2/3 signaling, and consequently FoxP3 expression [254]. Additionally, proteasomal FoxP3 degradation is inhibited by zinc due to histone deacetylase Sirtuin-1 (Sirt1) inhibition [255]. Other molecular targets, the interferon regulatory factor (IRF)-1 and Krüppel-like-factor 10 (KLF-10), are able to modulate FoxP3 activity, thereby influencing Treg cell development [256]. IRF-1 is referred to as a negative regulator of FoxP3 expression, as in vivo IRF-1 deficiency results in a selective and marked increase in highly differentiated and activated Treg cells [257]. KLF-10 is an essential transcription factor for proper Treg cell function, since KLF-10 deficiency leads to impaired cell differentiation, skewed cytokine profiles with enhanced Th1, Th2, and Th17 cytokines, and a reduced capacity for suppression of effector cells [258]. Zinc supplementation dampened IRF-1 activity and induced KLF-10 expression, leading to strengthened pro-tolerogenic immune response [256].

During zinc deficiency, studies reported an imbalance between Th1 and Th2 cell differentiation, resulting in a diminished Th1 immune response with lowered IFN-γ and IL-2 production, whereas the Th2 cytokines (IL-4, IL-6, and IL-10) remain unaffected [249,259,260]. In contrast, it was reported that zinc promotes a Th1 immune response by augmenting the gene expression of IFN-γ and IL-2 [261].

IL-2 is known to induce proliferation and development of effector functions in T cells, inducing antiapoptotic and cell-cycle-related genes as well as certain cytokines and lineage decisive factors [262].

When IL-2 binds to the IL-2 receptor, signal transduction via Jak1 and Jak3 activates numerous signaling pathways, including PI3K/Akt and MEK1/2-ERK1/2. Similar to IL-1 binding to its receptor, binding of IL-2 provokes an intracellular zinc level increase. Zinc is released from lysosomes (zincosomes), which is linked to ERK1/2 and Akt activation. In general, Akt activation is mediated by PI3K-produced PI(3,4,5)P3, which is degraded by phosphatase and tensin homologue deleted on chromosome 10 (PTEN). Interestingly, the mean inhibitory constant (IC50) of zinc for PTEN was found to be 0.59 nM, which would allow its blockade by intracellular zinc level, explaining the activation of Akt [263,264]. Similar results were uncovered for healthy men having a restricted zinc diet for 10 weeks, resulting in reduced IL-2R production [265]. In parallel, studies uncovered the transcription factor cAMP-responsive element modulator alpha (CREMα) to be upregulated due to zinc deficiency, leading to dampened IL-2 production [266,267].

The complexity of zinc’s effect on cell development and activation is highly dependent on environment, cellular status, and experimental setup, which may provide a possible explanation for different outcomes observed in the above-mentioned studies. In summary, these results show opposing effects of zinc deficiency and zinc supply on inflammatory and anti-inflammatory signaling pathways, underlying the importance of zinc for proper immune function.

## 7. Membrane Barrier Function and Zinc

One vital mechanism to defend invading pathogens are membrane barriers, such as those of the human skin or mucosa. The latter is particularly important in areas such as the lung and intestine, which are constantly exposed to myriads of pathogens and harmful substances. Zinc deficiency is associated with severe consequences, such as critically impaired intestinal and/or pulmonary health. Those tissues have a high turnover rate and are therefore the first to be affected by zinc deficiency, as indicated by morphological changes [268,269]; reduced self-renewal; severe degeneration of the intestinal epithelium [270,271]; reduction in integrity, resulting in increased membrane permeability [272]; and altered cell function, illustrated by impaired activity of brush border enzymes [273]. Studies even indicate disturbed production of the gastrointestinal mucus layer, which normally covers the whole gastrointestinal tract (GIT) and serves as an additional physical barrier for the underlying epithelium, protecting it against chemical and physical damage and pathogens [274]. Moreover, it serves as habitat for numerous commensal bacteria [275], and is important for nutritional absorption of macro- as well as micronutrients, such as zinc [276]. During zinc deficiency, mucus glycosylation is disordered, making the epithelium more vulnerable against pathogens, leading to overall degeneration of the mucosa that increases the occurrence of intestinal infections such as diarrhea [277]. Hence, zinc supplementation is recommended in acute diarrhea [117].

In general, zinc supply is reported to regenerate the intestinal epithelium, improve the absorption of water and electrolytes, increase levels of brush border enzymes, and allow better clearance of pathogens by modulating the immune system [273,278,279]. In detail, zinc inhibits cyclic adenosine monophosphate (cAMP)-induced chloride-dependent fluid secretion in enterocytes. Increased cAMP or cGMP levels mediate translocation of additional chloride channels to secrete more ions than usual. Due to osmotic reasons, water follows the salt concentration, making the stool more liquid, leading to diarrhea. Zinc supplementation results in a substantial reduction in cholera toxin (CT)-induced ion secretion and cAMP concentration [280]. On the molecular level, studies reveal structural proteins of the epithelium barrier to be influenced by zinc deficiency. Cellular connection of adjacent cells is facilitated by complexes called tight junctions (TJ) and adherens junctions (AJ). Important tight junction proteins and transmembrane proteins, such as ZO-1 and Claudin-1 or E-cadherin, are degraded during zinc deficiency, leading to increased leakage across the intestinal and pulmonal mucosa [272,281,282]. Thus, neutrophil leukocytes can migrate through the disrupted mucosa, causing acute and/or chronic infection that eventually contributes to intestinal and lung diseases.

Zinc supplementation is able to restore membrane function and structure [279], since it acts as a stabilizer of cell membranes and as essential cofactor in transcription factors and enzymes. For instance, zinc gives resistance to epithelial apoptosis through cytoprotection against reactive oxygen species and bacterial toxins, possibly through antioxidant activity of the cysteine-rich metallothioneins [283]. Investigations in patients suffering from *Shigella* infection showed that zinc supplementation improves the intestinal mucosal permeability, alkaline phosphatase activity, and nitrogen absorption [184]. Furthermore, lymphocyte proliferation is augmented in response to phytohemagglutinin activation, and antigen-specific IgG titers are increased [185]. In addition, elevated serum antibody titers, B cell count, and plasma cell counts are observed [186]. These effects indicate an immune modulatory effect of zinc supplementation.

Studies on *Mycobacterium* infections unravel lower severity of erythema nodosum, reduced infiltration and bacterial index of granuloma, earlier sputum conversion, and resolution of X-ray lesion areas in response to zinc supplementation (see Table 2) [170,181]. Effective clearance of mycobacterial infections requires a Th1-mediated activation of infected macrophages by IFN-γ [284]. Zinc improves the imbalance in T-cell subpopulations, as described earlier.

Moreover, zinc supplementation reduced the incidence of respiratory tract infections and pneumonia in children in developing countries and in nursing home residents [285,286]. For instance, the recently upcoming COVID-19 infection predominantly affects the respiratory system, resulting in pneumonia and acute respiratory distress syndrome [287], which often leads to the requirement of mechanical ventilation [288] and/or to long COVID symptoms [289]. Zinc supplementation, especially in combination with zinc ionophore pyrithione, has been shown to inhibit SARS-coronavirus RNA polymerase (RNA dependent RNA polymerase, RdRp) activity by decreasing its replication [290]. Moreover, ciliary length and ciliary beat frequency were improved in bronchial epithelium in vitro [291,292]. Another zinc-dependent effect is the upregulation of antiviral enzymes, such as latent ribonuclease (RNaseL) and protein kinase RNA-activated (PKR), which are involved in viral RNA degradation and inhibition of viral RNA translation [293]. In addition, viral entry into the cell, replication, and viral protein translation are known to be altered by zinc. This was especially shown for Picornaviridae causing common cold [97,294]. The effects of zinc treatment in common cold patients are summarized in Table 2. In respiratory syncytial virus (RSV) infection, viral replication and RSV plaque formation was found to be more than 1000-fold reduced due to zinc treatment, while during zinc deficiency, higher susceptibility to pneumonia was found [295,296]. Regarding COVID-19 infection, numerous clinical trials have been registered for COVID-19 infection [15], and some, but not all, show beneficial effects on hospitalization rates or severity of infection, as listed in Table 2.

Taken together, the above-described effects of zinc supply underscore the fundamental role of the micronutrient zinc in a healthy immune response. There is a great potential to improve the use of zinc as therapeutic agent in various diseases. However, the interactions between micronutrients need further study to develop supplementation measures that target multiple deficiencies to successfully support disease treatment.

## Figures and Tables

**Figure 1 biomolecules-12-01748-f001:**
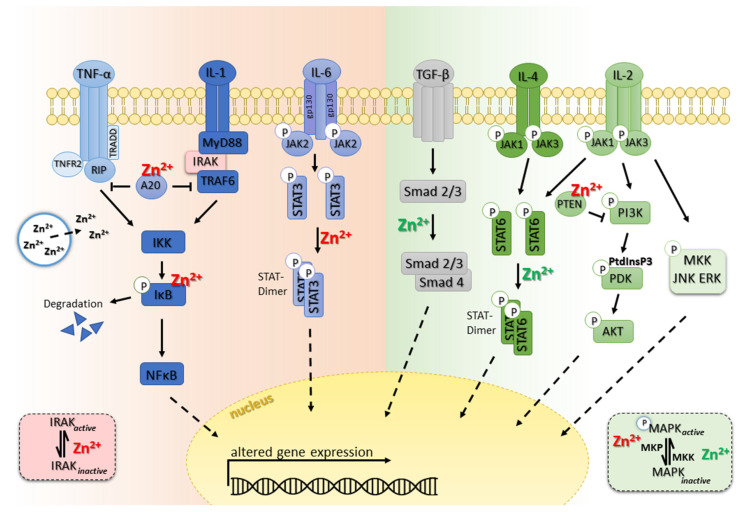
Inhibitory (red) and activating (green) effects of zinc supply on human pro- and anti-inflammatory signaling pathways.

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
