# Peer review of "Zinc in Human Health and Infectious Diseases"

_biomolecules, 2022, doi:10.3390/biom12121748_

Round 1
Reviewer 1 Report
The manuscript entitled "Zinc in human health and infectious diseases" is interesting and useful as it allows to deepen the knowledge of the current molecular mechanisms underlying the development of a pro- and anti-inflammatory immune response in the cases of zinc deficiency and zinc supplementation.
However, there are a few minor points that I wonder about, and I would like to make authors add or improve the following points:
1. Please add the aims of this work at the end of the introduction
2. Please briefly describe the searching methodology and inclusion/exclusion criteria as well as the timeline
3. How can we assess zinc status in humans? I think that a short paragraph with a description of diagnostics methods will be appreciated
Author Response
Point-by-Point Reply to Reviewer 1
Reviewer #1: The manuscript entitled "Zinc in human health and infectious diseases" is interesting and useful as it allows to deepen the knowledge of the current molecular mechanisms underlying the development of a pro- and anti-inflammatory immune response in the cases of zinc deficiency and zinc supplementation.
We thank the reviewer for its expert reviewing and for the positive evaluation. We changed the manuscript in all minor points requested by the reviewer and hope it fits its expectations. We agree that the manuscript gained significantly in its significance by this changes. All changes are marked in red.
Minor points
- Please add the aims of this work at the end of the introduction
We agree to the reviewer that this is helpful for the reader. The aim of this work was added as second last paragraph at the end of the introduction.
- Please briefly describe the searching methodology and inclusion/exclusion criteria as well as the timeline
We agree to the reviewer that this information is important, although this is not a systematic review. The search strategy was added as last paragraph at the end of the introduction.
- How can we assess zinc status in humans? I think that a short paragraph with a description of diagnostics methods will be appreciated
We agree to the reviewer that this is a very important point, although this is the most important and unsolved problem in clinical zinc research. We added a new paragraph “Assessment of Zinc Status” to the manuscript.
Reviewer 2 Report
1. Please briefly describe Zincosome where it is mentioned with some further details and how zincosomes are related to lysosome or if they are specialized lysosome. Describe it briefly with reference (line#576).
2. In heading no. 3, i.e., “Regulation of zinc homeostasis”, clearly explain the types of zinc transporters as are ‘Zrt’ and ‘Irt’ the sub types of SLC39As family? And what are ZIP? Is Zip also any subtype or is used for the previously mentioned subtypes. This is not clear in the sentence (line # 166 and 167)
3. In line # 229-231, explain how expression patterns and different functions suggests strong interplay between MT and the immune system, which types of pattern or relation, describe briefly. Or write it where details are givens.
4. Few grammatical mistakes, Authors need to proofread the manuscript.
Author Response
Point-by-Point Reply to Reviewer 2
We thank the reviewer for its expert reviewing and for the positive evaluation. We changed the manuscript in all minor points requested by the reviewer and hope it fits its expectations. We agree that the manuscript gained significantly in its significance by this changes. All changes are marked in red.
Reviewer #2:
- Please briefly describe Zincosome where it is mentioned with some further details and how zincosomes are related to lysosome or if they are specialized lysosome. Describe it briefly with reference (line#576).
We agree to the reviewer that this is helpful for the reader. The information on zincosomes is given in lines 114-118..
- In heading no. 3, i.e., “Regulation of zinc homeostasis”, clearly explain the types of zinc transporters as are ‘Zrt’ and ‘Irt’ the sub types of SLC39As family? And what are ZIP? Is Zip also any subtype or is used for the previously mentioned subtypes. This is not clear in the sentence (line # 166 and 167)
We agree to the reviewer that this is helpful for the reader. The requested information is given in lines 206-207.
- In line # 229-231, explain how expression patterns and different functions suggests strong interplay between MT and the immune system, which types of pattern or relation, describe briefly. Or write it where details are givens.
We agree to the reviewer that this is helpful for the reader. The requested information is given in lines 251-267.
- Few grammatical mistakes, Authors need to proofread the manuscript.
We corrected errors throughout the text.
